# Timing is Everything: Learning to Act Selectively with Costly Actions and Constraints

**David Mguni**[†,1]**, Aivar Sootla**[1]**, Juliusz Ziomek**[1]**, Oliver Slumbers**[1,2]**, Zipeng Dai**[1]**,
Kun Shao**[1]**, Jun Wang**[1,2]
[1]Huawei Noah's Ark Lab
[2]UCL, London, United Kingdom

## Abstract

Many real-world settings involve costs for performing actions; transaction costs in financial systems and fuel costs being common examples. In these settings, performing actions at each time step quickly accumulates costs leading to vastly suboptimal outcomes. Additionally, repeatedly acting produces wear and tear and ultimately, damage. Determining *when to act* is crucial for achieving successful outcomes and yet, the challenge of efficiently *learning* to behave optimally when actions incur minimally bounded costs remains unresolved. In this paper, we introduce a reinforcement learning (RL) framework named **L**earnable **I**mpulse **C**ontrol **R**einforcement **A**lgorithm (LICRA), for learning to optimally select both when to act and which actions to take when actions incur costs. At the core of LICRA is a nested structure that combines RL and a form of policy known as *impulse control* which learns to maximise objectives when actions incur costs. We prove that LICRA, which seamlessly adopts any RL method, converges to policies that optimally select when to perform actions and their optimal magnitudes. We then augment LICRA to handle problems in which the agent can perform at most $k < \infty$ actions and more generally, faces a budget constraint. We show LICRA learns the optimal value function and ensures budget constraints are satisfied almost surely. We demonstrate empirically LICRA's superior performance against benchmark RL methods in OpenAI gym's *Lunar Lander* and in *Highway* environments and a variant of the Merton portfolio problem within finance.

## 1 Introduction

There are many settings in which agents incur costs each time they perform an action. Transaction costs in financial settings (Mguni, 2018a), fuel expenditure (Zhao et al., 2018), toxicity as a side effect of controlling bacteria (Sootla et al., 2016) and physical damage produced by repeated action that produces wear and tear are just a few among many examples (Grandt Jr, 2011). In these settings, performing actions at each time step is vastly suboptimal since acting in this way results in prohibitively high costs and undermines the service life of machinery. Minimising wear and tear is an essential attribute to safeguard against failures that can result in catastrophic losses (Grandt Jr, 2011).

Reinforcement learning (RL) is a framework that enables autonomous agents to learn complex behaviours from interactions with the environment (Sutton & Barto, 2018). Within the standard RL paradigm, determining optimal actions involves making a selection among many (possibly infinite) actions; a procedure that must be performed at each time-step as the agent decides on an action. In unknown settings, the agent cannot immediately exploit any topological structure of the action set. Consequently, learning *not to take* an action i.e performing a zero or *null action*, involves expensive optimisation procedures over the entire action set. Since this must be done at each state, this process is vastly inefficient for learning optimal policies when the agent incurs costs for acting.

In this paper, we tackle this problem by developing an RL framework for finding both an optimal criterion to determine whether or not to execute actions as well as learning optimal actions. A key component of our framework is a novel combination of RL with a form of policy known as *impulse control* (Mguni, 2018d;a). This enables the agent to determine the appropriate points to perform an action as well as the optimal action itself. Despite its fundamental importance as a tool for tackling

decision problems with costly actions (Korn, 1999; Mitchell et al., 2014), presently, the use of impulse control within learning contexts (and unknown environments) is unaddressed.

We present an RL impulse control framework called LICRA, which, to our knowledge, is the first learning framework for impulse control. To enable learning optimal impulse control policies in unknown environments, we devise a framework that consists of separate RL components for learning when to act and, how to act, optimally. The resulting framework is a structured two-part learning process which differs from current RL protocols. In LICRA, at each time step, the agent firstly makes a decision whether to act or not leading to a binary decision space $\{0, 1\}$. The second decision part determines the best action to take. This generates a subdivision of the state space into two regions; one in which the agent performs actions and another in which it does not act at all. We show the set of states where the agent ought to act is characterised using an easy-to-evaluate criterion on the value function and, LICRA quickly learns this set within which it then learns which actions are optimal.

We then establish theory that ensures convergence of a Q-learning variant of LICRA to the optimal policy for such settings. To do this, we give a series of results namely:

**i)** We establish a dynamic programming principle (DPP) for impulse control and show that the optimal value function can be obtained as a limit of a value iterative procedure (Theorem 1) which lays the foundation for an RL approach to impulse control.
**ii)** We extend result i) to a new variant of Q-learning which enables the impulse control problem to be solved using our RL method (Theorem 2). Thereafter, we extend the result to (linear) function approximators enabling the value function to be parameterised (Theorem 3).
**iii)** We characterise the optimal conditions for performing an action which we reveal to be a simple 'obstacle condition' involving the agent's value function (Proposition 1). Using this, the agent can quickly determine whether or not it should act and if so, then learn what the optimal action is.
**iv)** In Sec. 6, we extend LICRA to include budgetary constraints so that each action draws from a fixed budget which the agent must stay within. Analogous to the development of i), we establish another DPP from which we derive a Q-learning variant for tackling impulse control with budgetary constraints (Theorem 4). A particular case of a budget constraint is when the number of actions the agent can take over the horizon is capped.
Lastly, we perform a set of experiments to validate our theory within the *Highway* driving simulator and OpenAI's *LunarLander* (Brockman et al., 2016).

As we demonstrate in our experiments, LICRA learns to compute the optimal problems in which the agent faces costs for acting in an efficient way which outperforms leading RL baselines. Second, as demonstrated in Sec. 6, LICRA handles settings in which the agent has a cap the total number of actions it is allowed to execute and more generally, generic budgetary constraints. LICRA is able to accommodate any RL algorithm unlike various RL methods designed to handle budgetary constraints.

## 2 RELATED WORK

In continuous-time optimal control theory (Øksendal, 2003), problems in which the agent faces a cost for each action are tackled with a form of policy known as *impulse control* (Mguni, 2018d;a; Al-Fagih, 2015). In impulse control frameworks, the dynamics of the system are modified through a sequence of discrete actions or *bursts* chosen at times that the agent chooses to apply the control policy. This distinguishes impulse control models from classical decision methods in which an agent takes actions at each time step while being tasked with the decision of only which action to take. Impulse control models represent appropriate modelling frameworks for financial environments with transaction costs, liquidity risks and economic environments in which players face fixed adjustment costs (e.g. *menu costs*) (Korn, 1999; Mguni, 2018b; Mundaca & Øksendal, 1998).

The current setting is intimately related to the *optimal stopping problem* which widely occurs in finance, economics and computer science (Mguni, 2019; Tsitsiklis & Van Roy, 1999). In the optimal stopping problem, the task is to determine a criterion that determines when to arrest the system and receive a terminal reward. In this case, standard RL methods are unsuitable since they require an expensive sweep (through the set of states) to determine the optimal point to arrest the system. The current problem can be viewed as an augmented problem of optimal stopping since the agent must now determine both a sequence of points to perform an action or *intervene* and their optimal magnitudes — only acting when the cost of action is justified (Øksendal & Sulem, 2019). Adapting

RL to tackle optimal stopping problems has been widely studied (Tsitsiklis & Van Roy, 1999; Becker et al., 2018; Chen et al., 2020) and applied to a variety of real-world settings within finance (Fathan & Delage, 2021) and network operating systems (Alcaraz et al., 2020). Our work serves as a natural extension of RL approaches to optimal stopping to the case in which the agent must decide at which points to take a series of actions. As with optimal stopping, standard RL methods cannot *efficiently* tackle this problem since determining whether to perform a 0 action requires a costly sweep through the action space at every state (Tsitsiklis & Van Roy, 1999). In (Pang et al., 2021) the authors introduce "sparse action" with a similar motivation as impulse control. However, the authors treat only the discrete action space case. Moreover, (Pang et al., 2021) do not provide a general theoretical framework of dealing with "sparse actions" but develop purely algorithmic solutions. Additionally, unlike the approach taken in (Pang et al., 2021), the problem setting we consider is one in which the agent faces a cost for each action - this produces a need for the agent to be selective about where it performs actions (but does not necessarily constrain the magnitude or choice of those actions).

Our framework bears resemblance to hierarchical RL (HRL) (Barto & Mahadevan, 2003). Policies in this hierarchy have usually different objectives. For example in sub-goal learning (Noelle, 2019), a top-level policy decides the goal of a low-level policy. In this case, the top-level policy gives control to the low-level policy for a number of time steps and/or until the sub-goal is achieved. In contrast, LICRA's top-level policy must make a decision at every time step. One can note further similarities with a specific HRL framework — RL with *options* (Precup et al., 1998; Klissarov & Precup, 2021). There the agent chooses an option and then selects an action (or a sequence of actions) based on the current state and the chosen option. Therefore, there are two policies that the agent learns. Options and HRL offer abstraction frameworks with higher levels of internal structure than most of the standard RL algorithms. This can significantly speed up learning in many situations. For example, in a setting with sparse rewards, a subgoal discovery algorithm can guide the agent to an optimal policy more quickly. Our impulse control framework offers an even higher level of structure than options and HRL in order to determine when to act (Jinnai et al., 2019). This is achieved by exploiting the specificity of the impulse control setting which includes a cost of acting while employing a structure that enables isolating null actions from the action space.

## 3 PRELIMINARIES

In RL, an agent sequentially selects actions to maximise its expected returns. The underlying problem is typically formalised as an MDP $\langle \mathcal{S}, \mathcal{A}, P, R, \gamma \rangle$ where $\mathcal{S} \subset \mathbb{R}^p$ is the set of states, $\mathcal{A} \subset \mathbb{R}^k$ is the (continuous) set of actions, $P : \mathcal{S} \times \mathcal{A} \times \mathcal{S} \to [0, 1]$ is a transition probability function describing the system's dynamics, $R : \mathcal{S} \times \mathcal{A} \to \mathbb{R}$ is the reward function measuring the agent's performance and the factor $\gamma \in [0, 1)$ specifies the degree to which the agent's rewards are discounted over time (Sutton & Barto, 2018). At time $t \in 0, 1, \ldots$, the system is in state $s_t \in \mathcal{S}$ and the agent must choose an action $a_t \in \mathcal{A}$ which transitions the system to a new state $s_{t+1} \sim P(\cdot|s_t, a_t)$ and produces a reward $R(s_t, a_t)$. A policy $\pi : \mathcal{S} \times \mathcal{A} \to [0, 1]$ is a probability distribution over state-action pairs where $\pi(a|s)$ represents the probability of selecting action $a \in \mathcal{A}$ in state $s \in \mathcal{S}$. The goal of an RL agent is to find a policy $\hat{\pi} \in \Pi$ that maximises its expected returns given by the value function: $v^\pi(s) = \mathbb{E}[\sum_{t=0}^\infty \gamma^t R(s_t, a_t)|a_t \sim \pi(\cdot|s_t), s_0 = s]$ where $\Pi$ is the agent's policy set. The action value function is given by $Q(s, a) = \mathbb{E}[\sum_{t=0}^\infty R(s_t, a_t)|a_0 = a, s_0 = s]$.

We consider a setting in which the agent faces at least some minimal cost for each action it performs. With this, the agent's task is to maximise:

$$v^\pi(s) = \mathbb{E}\left[ \sum_{t=0}^\infty \gamma^t \left\{ \mathcal{R}(s_t, a_t) - \mathcal{C}(s_t, a_t) \right\} \Big| s_0 = s \right], \tag{1}$$

where for any state $s \in \mathcal{S}$ and any action $a \in \mathcal{A}$, the functions $\mathcal{R}$ and $\mathcal{C}$ are given by $\mathcal{R}(s, a) = R(s, a)\mathbf{1}_{a \in \mathcal{A}/\{0\}} + R(s, 0)(1 - \mathbf{1}_{a \in \mathcal{A}/\{0\}})$ where $\mathbf{1}_{a \in \mathcal{A}/\{0\}}$ is the indicator function which is 1 when $a \in \mathcal{A}/\{0\}$ and 0 otherwise and $\mathcal{C}(s, a) := c(s, a)\mathbf{1}_{a \in \mathcal{A}/\{0\}}$ where $c : \mathcal{S} \times \mathcal{A} \to \mathbb{R}$ is a strictly positive (cost) function that introduces a cost each time the agent performs an action. Examples of the cost function is a quasi-linear function of the form $c(s_t, a_t) = \kappa + f(a_t)$ where $f : \mathcal{A} \to \mathbb{R}_{>0}$ and $\kappa$ is a positive real-valued constant. Since acting at each time step would incur prohibitively high costs, the agent must be selective when to perform an action. Therefore, in this setting, the agent's problem is augmented to learning both an optimal policy for its actions and, learning at which states to apply

its action policy. Examples of problem settings of this kind include financial portfolio problems with transaction costs (Davis & Norman, 1990) where each investment incurs a fixed minimal cost and robotics where actions produce mechanical stress leading to fatigue (Grandt Jr, 2011).

## 4 THE LICRA FRAMEWORK

In RL, the agent's problem involves learning to act at *every* state including those in which actions do not significantly impact on its total return. While we can add a zero action to the action set $\mathcal{A}$ and apply standard methods, we argue that this may not be the best solution in many situations. We argue the optimal policy has the following form:

$$\widetilde{\pi}(\cdot|s) = \begin{cases} a_t & s \in \mathcal{S}_I, \\ 0 & s \notin \mathcal{S}_I, \end{cases} \tag{2}$$

which implies that we simplify policy learning by determining the set $\mathcal{S}_I$ first — the set where we actually need to learn the policy.

We now introduce a learning method for producing impulse controls which enables the agent to learn to select states to perform actions. Therefore, now agent is tasked with learning to act at states that are most important for maximising its total return given the presence of the cost for each action. To do this effectively, at each state the agent first makes a *binary decision* to decide to perform an action. Therefore, LICRA consists of two core components: first, an RL process $\mathfrak{g} : \mathcal{S} \times \{0, 1\} \to [0, 1]$ and a second RL process $\pi : \mathcal{S} \times \mathcal{A} \to [0, 1]$. The role of $\mathfrak{g}$ is to determine whether or not an action is to be performed by the policy $\pi$ at a given state $s$. In LICRA, the policy $\pi$ first proposes an action $a \in \mathcal{A}$ which is observed by the policy $\mathfrak{g}$. If activated, the policy $\pi$ determines the action to be selected. Therefore, $\mathfrak{g}$ prevents actions under $\pi$ for which the change in expected return does not exceed the costs incurred for taking such actions. While we consider now two policies $\pi, \mathfrak{g}$, the cardinality of the action space does not change. Crucially, the agent must compute optimal actions at only the subset of states chosen by $\mathfrak{g}$. By isolating the decision of whether to act or not, the LICRA framework may also reduce the computational complexity in this setting.

---

**Algorithm 1: L**earnable **I**mpulse **C**ontrol **R**einforcement **A**lgorithm (LICRA)

---

1: **Input:** Stepsize $\alpha$, batch size $B$, episodes $K$, steps per episode $T$, mini-epochs $e$
2: **Initialise:** Policy network (acting) $\pi$, Policy network (switching) $\mathfrak{g}$,
   Critic network (acting )$V_\pi$,Critic network (switching )$V_\mathfrak{g}$
3: Given reward objective function, $R$, initialise Rollout Buffers $\mathcal{B}_\pi, \mathcal{B}_\mathfrak{g}$ (use Replay Buffer for SAC)
4: **for** $N_{episodes}$ **do**
5:     Reset state $s_0$, Reset Rollout Buffers $\mathcal{B}_\pi, \mathcal{B}_\mathfrak{g}$
6:     **for** $t = 0, 1, \ldots$ **do**
7:         Sample $a_t \sim \pi(\cdot|s_t)$
8:         Sample $g_t \sim \mathfrak{g}(\cdot|s_t)$
9:         **if** $g_t = 1$ **then**
10:             Apply $a_t$ so $s_{t+1} \sim P(\cdot|a_t, s_t)$,
11:             Receive rewards $r_t = \mathcal{R}(s_t, a_t) - c(s_t, a_t)$
12:             Store $(s_t, a_t, s_{t+1}, r_t)$ in $\mathcal{B}_\pi$
13:         **else**
14:             Apply the null action so $s_{t+1} \sim P(\cdot|0, s_t)$,
15:             Receive rewards $r_t = \mathcal{R}(s_t, 0)$.
16:         **end if**
17:         Store $(s_t, g_t, s_{t+1}, r_t)$ in $\mathcal{B}_\mathfrak{g}$
18:     **end for**
19:     **// Learn the individual policies**
20:     Update policy $\pi$ and critic $V_\pi$ networks using $\mathfrak{B}_\pi$
21:     Update policy $\mathfrak{g}$ and critic $V_\mathfrak{g}$ networks using $\mathfrak{B}_\mathfrak{g}$
22: **end for**

---

LICRA consists of two independent procedures: a learning process for the policy $\pi$ and simultaneously, a learning process for the impulse policy $\mathfrak{g}$ which determines at which states to perform an

action. In Sec. 5, we prove the convergence properties of LICRA. In our implementation, we used proximal policy optimisation (PPO) (Schulman et al., 2017) for the policy $\pi$ and for the impulse policy $\mathfrak{g}$, whose action set consists of two actions (intervene or do not intervene) we used a soft actor critic (SAC) process (Haarnoja et al., 2018). LICRA is a plug & play framework which enables these RL components to be replaced with any RL algorithm of choice. Above is the pseudocode for LICRA, we provide full details of the code in Sec. 9 of the Appendix.

## 5    CONVERGENCE AND OPTIMALITY OF LICRA WITH Q-LEARNING

A key aspect of our framework is the presence of two RL processes that make decisions in a sequential order. In order to determine when to act the policy $\mathfrak{g}$ must learn the states to allow the policy $\pi$ to perform an action which the policy $\pi$ must learn to select optimal actions whenever it is allowed to execute an action. In this section, we prove that LICRA converges to an optimal solution of the system. Our proof is instantiated in a Q-learning variant of LICRA which serves as the natural basis for other extensions such as actor-critic methods. We then extend the result to allow for (linear) function approximators. We provide a result that shows the optimal intervention times are characterised by an 'obstacle condition' which can be evaluated online therefore allowing the $\mathfrak{g}$ policy to be computed online.

Given a function $Q : \mathcal{S} \times \mathcal{A} \to \mathbb{R}$, $\forall \pi, \pi' \in \Pi$ and $\forall \mathfrak{g}, \forall s_{\tau_k} \in \mathcal{S}$, we define the intervention operator $\mathcal{M}$ by $\mathcal{M}^\pi[Q^{\pi',\mathfrak{g}}(s_{\tau_k}, a)] := \mathcal{R}(s_{\tau_k}, a_{\tau_k}) - c(s_{\tau_k}, a_{\tau_k}) + \gamma \sum_{s' \in \mathcal{S}} P(s'; a_{\tau_k}, s) Q^{\pi',\mathfrak{g}}(s', a_{\tau_k}) | a_{\tau_k} \sim \pi(\cdot | s_{\tau_k})$.[1] The interpretation of $\mathcal{M}^{\pi'}[Q^{\pi,\mathfrak{g}}]$ is the following: suppose that at time $\tau_k$ the system is at a state $s_{\tau_k}$ and the agent performs an immediate action $a_{\tau_k} \sim \pi'(\cdot | s_{\tau_k})$. A cost of $c(s_{\tau_k}, a_{\tau_k})$ is then incurred by the agent and the system transitions to $s' \sim P(\cdot; a_{\tau_k}, s_{\tau_k})$ whereafter the agent executes the policy $(\pi, \mathfrak{g})$. Therefore $\mathcal{M}^{\pi'} Q^{\pi,\mathfrak{g}}$ is the expected future stream of rewards after an immediate action minus the cost of action. This object plays a crucial role in the LICRA framework which as we later discuss, exploits the cost structure of the problem to determine when the agent should take an action. Denote by $\mathcal{M}[Q^{\pi,\mathfrak{g}}]$ the intervention operator acting on $Q^{\pi,\mathfrak{g}}$ when the immediate action is chosen according to an epsilon-greedy policy. Given any $v^{\pi,\mathfrak{g}} : \mathcal{S} \to \mathbb{R}$, the Bellman operator $T$ is:

$$T v^{\pi,\mathfrak{g}}(s) := \max \left\{ \mathcal{M}[Q^{\pi,\mathfrak{g}}(s,a)], \mathcal{R}(s,0) + \gamma \sum_{s' \in \mathcal{S}} P(s'; 0, s) v^{\pi,\mathfrak{g}}(s') \right\}, \qquad \forall s \in \mathcal{S}. \quad (3)$$

The Bellman operator captures the nested sequential structure of the LICRA method. In particular, the structure in (3) consists of an inner structure that consists of two terms: the first term $\mathcal{M}[Q^{\pi,\mathfrak{g}}]$ is the estimated expected return given the current best estimate of the optimal action is executed and the policy $\pi$ is executed thereafter. The second term $\mathcal{R}(s,0) + \gamma \sum_{s' \in \mathcal{S}} P(s', 0, s) v^{\pi,\mathfrak{g}}(s')$ isolates the estimated expected return following no action and when the policy $\pi$ is executed thereafter. Lastly, the outer structure is an optimisation which compares the two quantities and selects the maximum. Note that when $\mathcal{A}$ is a continuous (dense) set, this reduces the burden on an optimisation (possibly over a function approximator) to extract the point $0$ in the action space when it is optimal not to act.

Our first result proves $T$ is a contraction operator in particular, the following bound holds:

**Lemma 1** *The Bellman operator $T$ is a contraction, that is the following bound holds:*

$$\| T v - T v' \| \le \gamma \| v - v' \|,$$

*where $v, v'$ are elements of a finite normed vector space. We can now state our first main result:*

**Theorem 1** *Given any $v^{\pi,\mathfrak{g}} : \mathcal{S} \times \mathcal{A} \to \mathbb{R}$, the optimal value function is given by $\lim_{k \to \infty} T^k v^{\pi,\mathfrak{g}} = \max_{\hat{\pi}, \hat{g} \in \Pi} v^{\hat{\pi}, \hat{g}} = v^{\pi^\star, g^\star}$ where $(\pi^\star, g^\star)$ is the optimal policy pair.*

The result of Theorem 1 enables the solution to the agent's impulse control problem to be determined using a value iteration procedure. Moreover, Theorem 1 enables a Q-learning approach (Bertsekas, 2012) for finding the solution to the agent's problem.

---

[1] Note for the term $\mathcal{M}Q$, the action input of the $Q$ function is a dummy variable decided by the operator $\mathcal{M}$.

**Theorem 2** *Consider the following Q-learning variant:*

$$Q_{t+1}(s_t, a_t) = Q_t(s_t, a_t)$$
$$+ \alpha_t(s_t, a_t) \left[ \max \left\{ \mathcal{M}[Q_t(s_t, a_t)], \mathcal{R}(s_t, 0) + \gamma \sup_{a' \in \mathcal{A}} Q_t(s_{t+1}, a') \right\} - Q_t(s_t, a_t) \right], \quad (4)$$

*then $Q_t$ converges to $Q^\star$ with probability 1, where $s_t, s_{t+1} \in \mathcal{S}$ and $a_t \in \mathcal{A}$.*

We now extend the result to (linear) function approximators:

**Theorem 3** *Given a set of linearly independent basis functions $\Phi = \{\phi_1, \dots, \phi_p\}$ with $\phi_k \in L_2, \forall k$. LICRA converges to a limit point $r^\star \in \mathbb{R}^p$ which is the unique solution to $\Pi \mathfrak{F}(\Phi r^\star) = \Phi r^\star$ where $\mathfrak{F}Q := \mathcal{R} + \gamma P \max\{\mathcal{M}[Q], Q\}$ and $r^\star$ satisfies: $\|\Phi r^\star - Q^\star\| \leq (1 - \gamma^2)^{-1/2} \|\Pi Q^\star - Q^\star\|$.*

The theorem establishes the convergence of the LICRA Q-learning variant to a stable point with the use of linear function approximators. The second statement bounds the proximity of the convergence point by the smallest approximation error that can be achieved given the choice of basis functions.

Having constructed a procedure to find the optimal agent's optimal value function, we now seek to determine the conditions when an intervention should be performed. Let us denote by $\{\tau_k\}_{k \geq 0}$ the points at which the agent decides to act or *intervention times*, so for example if the agent chooses to perform an action at state $s_6$ and again at state $s_8$, then $\tau_1 = 6$ and $\tau_2 = 8$. The following result characterises the optimal intervention policy $\mathfrak{g}$ and the optimal times $\{\tau_k\}_{k \geq 0}$.

**Proposition 1** *The policy $\mathfrak{g}$ is given by: $\mathfrak{g}(s_t) = H(\mathcal{M}^\pi[Q^{\pi,\mathfrak{g}}] - Q^{\pi,\mathfrak{g}})(s_t, a_t), \quad \forall s_t \in \mathcal{S}$, where $Q^{\pi,\mathfrak{g}}$ is the solution in Theorem 1, $\mathcal{M}$ is the intervention operator and $H$ is the Heaviside function, moreover the intervention times are $\tau_k = \inf\{\tau > \tau_{k-1} | \mathcal{M}^\pi[Q^{\pi,\mathfrak{g}}] = Q^{\pi,\mathfrak{g}}\}$.*

Prop. 1 characterises the (categorical) distribution $\mathfrak{g}$. Moreover, given the function $Q$, the times $\{\tau_k\}$ can be determined by evaluating if $\mathcal{M}[Q] = Q$ holds. A key aspect of Prop. 1 is that it exploits the cost structure of the problem to determine when the agent should perform an intervention. In particular, the equality $\mathcal{M}[Q] = Q$ implies that performing an action is optimal.

## 6 BUDGET AUGMENTED LICRA VIA STATE AUGMENTATION

We now tackle the problem of RL with a budget. To do this, we combine the above impulse control technology with state augmentation technique proposed in (Sootla et al., 2022) The mathematical formulation of the problem is now given by the following for any $s \in \mathcal{S}$:

$$\max_{\pi \in \Pi, g} v^{\pi,\mathfrak{g}}(s) \text{ s. t. } n - \sum_{t=0}^{\infty} \sum_{k \geq 1} \delta_{\tau_k}^t \geq 0, \quad (5)$$

where $n \in \mathbb{N}$ is a fixed value that represents the maximum number of allowed interventions and $\sum_{k \geq 1} \delta_{\tau_k}^t$ is equal to one if an impulse was applied at time $t$ and zero if it was not. In order to avoid dealing with a constrained MDP, we propose to introduce a new variable $z_t$ tracking the remaining number of impulses: $z_t = n - \sum_{i=0}^{t-1} \sum_{k \geq 1} \delta_{\tau_k}^i$. We treat $z_t$ as another state and augment the state-space resulting in the transition $\widetilde{\mathcal{P}}$:

$$s_{t+1} \sim P(\cdot | s_t, a_t), \qquad z_{t+1} = z_t - \sum_{k \geq 1} \delta_{\tau_k}^t, \quad z_0 = n. \quad (6)$$

To avoid violations, we reshape the reward as follows: $\widetilde{\mathcal{R}}(s_t, z_t, a_t) = \begin{cases} \mathcal{R}(s_t, a_t) & z_t \geq 0, \\ -\Delta & z_t < 0, \end{cases}$ where $\Delta > 0$ is a large enough hyper-parameter ensuring there are no safety violations. To summarise, we aim to solve the following budgeted impulse control (BIC) problem $v^{\pi,\mathfrak{g}}(s, z) = \mathbb{E}\left[\sum_{t=0}^{\infty} \gamma^t \widetilde{\mathcal{R}}(s_t, z_t, a_t) | a_t \sim \pi(\cdot | s_t, z_t)\right]$, where the policy now depends on the variable $z_t$. Note $\widetilde{\mathcal{P}}$ in Equation 6 is a Markov process and, the rewards $\widetilde{\mathcal{R}}$ are bounded given the boundedness of $\mathcal{R}$. Therefore, we can apply directly the results for impulse control. We denote the augmented MDP by $\widetilde{M} = \langle \mathcal{S} \times \mathcal{Z}, \mathcal{A}, \widetilde{\mathcal{P}}, \widetilde{R}, \gamma \rangle$, where $\mathcal{Z}$ is the space of the augmented state. We have the following:

**Theorem 4** *Consider the MDP $\widetilde{M}$ for the BIC problem, then:*

*a) The Bellman equation holds, i.e.* $\exists \tilde{v}^{*,\boldsymbol{\pi},\mathfrak{g}} : \mathcal{S} \times \mathcal{Z} \rightarrow \mathbb{R}$ *such that* $\tilde{v}^{*,\boldsymbol{\pi},\mathfrak{g}}(s,z) = \max_{\boldsymbol{a} \in \mathcal{A}} \left( \widetilde{\mathcal{R}}(s,z,\boldsymbol{a}) + \gamma \mathbb{E}_{s',z' \sim \mathcal{P}} \left[ \tilde{v}^{*,\boldsymbol{\pi},\mathfrak{g}}(s',z') \right] \right)$, *where the optimal policy for $\widetilde{M}$ has the form* $\pi^*(\cdot|s,z)$; *b) Given a $\tilde{v} : \mathcal{S} \times \mathcal{Z} \rightarrow \mathbb{R}$, the stable point solution for $\widetilde{\mathcal{M}}$ is given by* $\lim_{k \to \infty} \tilde{T}^k \tilde{v}^{\boldsymbol{\pi},g} = \max_{\hat{\boldsymbol{\pi}} \in \boldsymbol{\Pi},\hat{\mathfrak{g}}} \tilde{v}^{\hat{\boldsymbol{\pi}},\hat{\mathfrak{g}}} = \tilde{v}^{\boldsymbol{\pi}^*,\mathfrak{g}^*}$, *where $(\boldsymbol{\pi}^*, \mathfrak{g}^*)$ is an optimal policy and $\tilde{T}$ is the Bellman operator of $\widetilde{M}$.*

The result has several important implications. The first is that we can use a modified version of LICRA to obtain the solution of the problem while guaranteeing convergence (under standard assumptions). Secondly, our state augmentation procedure admits a Markovian representation of the optimal policy.

## 7 EXPERIMENTS

We will now study empirically the performance of the LICRA framework. In experiments, we use different instances of LICRA, one where both policies are trained using PPO update (referred to as **LICRA_PPO**) and one where the policy deciding whether to act is trained using SAC and the other policy trained with PPO (referred to as **LICRA_SAC**). We have benchmarked both of these algorithms together with common baselines on environments, where it would be natural to introduce the concept of the cost associated with actions. We performed a series of ablation studies (see Appendix) which examine **i)** LICRA's ability to handle different cost functions including the case when $c(s,a) \equiv 0$. **ii)** the benefits of LICRA in settings with a varying intervention regions. **iii)** the performance of the LICRA Q-learning variant in a discrete state space environment. In the following experiments, all algorithms apply actions that are drawn from an action set which contains a $0$ action. When this action is executed it produces costs and exerts no influence on the transition dynamics. **Merton's Portfolio Problem with Transaction Costs.** To exemplify the problem in an applied

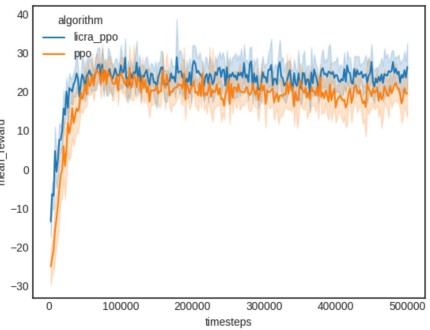

Figure 1: Training results in Merton investment problem for PPO style algorithms. (Confidence intervals over 20 seeds). Average final performance was 26.40 for LICRA_PPO and 19.60 for PPO.

setting, we choose a well-known problem in finance known as the Merton Investment Problem with transaction costs (Davis & Norman, 1990). In this simple environment, the agent can decide to move its wealth between a risky asset and a risk-free asset. The agent receives a reward only at the final step, equal to the utility of the portfolio with a risk aversion factor equal to 0.5. If the final wealth of risky asset is $s_T$ and final wealth of risk-free asset is $c_T$, then the agent will receive a reward of $u(x) = 2\sqrt{s_T + c_T}$. The wealth evolves according to the following SDE:

$$dW_t = (r + p_t(\mu - r))W_t + W_t p_t \sigma dB_t \tag{7}$$

where $W_t$ is the current wealth (the state variable), $dB_t$ is an increment of Brownian motion and $p_t$ is the proportion of wealth invested in the risky asset. We set the risk-free return $r = 0.01$, risky asset return $\mu = 0.05$ and volatility $\sigma = 1$. We discretise the action space so that at each step the agent has three actions available: move 10% of risky asset wealth to the risk-free asset, move 10% of risk-free asset wealth to the risky asset or do nothing. Each time the agent moves the assets, it incurs a cost of 1 i.e. a transaction fee. The agent can act after a time interval of 0.01 seconds and the episode ends after 75 steps. The results of training are shown in Fig. 1 which clearly demonstrates that LICRA_PPO learns faster than standard PPO.

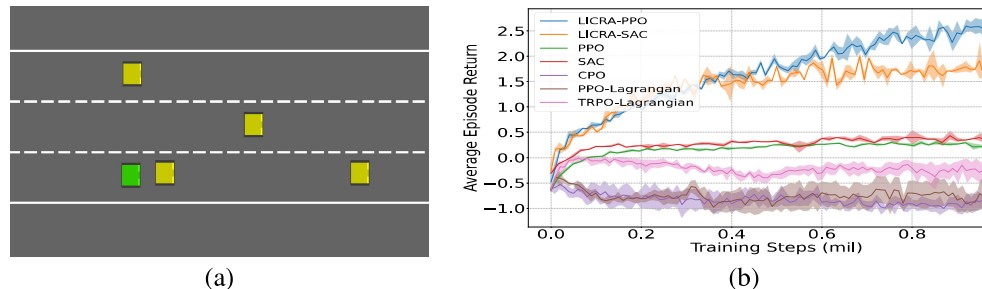

(a)                                      (b)

Figure 2: a) Drive Environment. b) Training results in drive environment. (Confidence intervals over 5 seeds) Average final performance was 1.75 for LICRA_SAC, 0.37 for SAC, 2.63 for LICRA_PPO, 0.24 for PPO.

**Driving Environment Fuel Rationing.** We studied an autonomous driving scenario where fuel-efficient driving is a priority. One of the main components of fuel-efficient driving is controlled usage of acceleration and braking, in the sense that 1) the amount of acceleration and braking should be limited 2) if accelerations should be slowly and gently. We believe this is a problem where LICRA should thrive as the impulse control agent can learn to restrict the amount of acceleration and braking in the presence of other cars and choose when to allow the car to decelerate naturally. We used the highway-env (Leurent, 2018) environment on a highway task (see Fig (2. a)) where the green vehicle is our controlled vehicle and the goal is to avoid crashing into other vehicles whilst driving at a reasonable speed. We add a cost function into the reward term dependent on the continuous acceleration action, $C(a_t) = K + a_t^2$, where $K > 0$ is a fixed constant cost of taking any action, and $a_t \in [-1, 1]$, with larger values of acceleration or braking being penalised more. The results are presented in Fig. (2.b). Notably, LICRA is able to massively outperform the baselines, especially our safety specific baselines which struggle to deal with the cost function associated with the environment. We believe one reason for the success of LICRA is that it is far easier for it to utilise the null action of zero acceleration/braking than the other algorithms, whilst all the algorithms have a guaranteed cost at every time step whilst not gaining a sizeable reward to counter the cost.

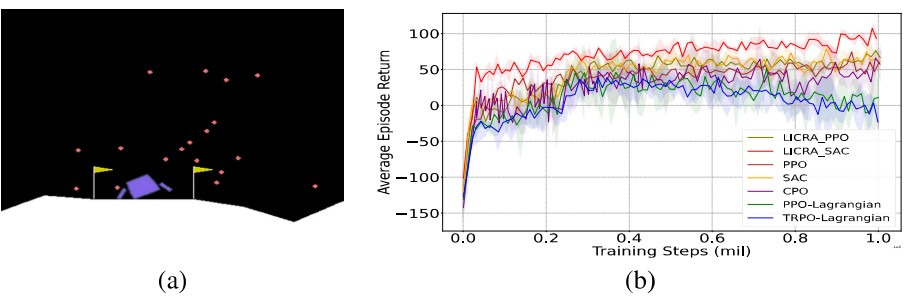

(a)                                      (b)

Figure 3: a) The lander must land on the pad between two flags. b) Training results in Lunar Lander. Note that we use 6 different random seeds to get the error bars in this figure. The average final performance was 98.03 for LICRA_SAC, 61.20 for SAC, 58.84 for LICRA_PPO, 49.77 for PPO.

**Lunar Lander Environment.** We tested the ability of LICRA to perform in environment that simulate real-world physical dynamics. We tested LICRA's performance the Lunar Lander environment in OpenAI gym (Brockman et al., 2016) which we adjusted to incorporate minimal bounded costs in the reward definition. In this environment, the agent is required to maintain both a good posture mid-air and reach the landing pad as quickly as possible. The reward function is given by:

$$\text{Reward}(s_t) = 3 * (1 - \mathbf{1}_{d_t - d_{t-1} = 0}) - 3 * (1 - \mathbf{1}_{v_t - v_{t-1} = 0}) - 3 * (1 - \mathbf{1}_{\omega_t - \omega_{t-1} = 0})$$

$$-0.03 * \text{FuelSpent}(s_t) - 10 * (v_t - v_{t-1}) - 10 * (\omega_t - \omega_{t-1}) + 100 * \text{hasLanded}(s_t)$$

where $d_t$ is the distance to the landing pad, $v_t$ is the velocity of the agent, and $\omega_t$ is the angular velocity of the agent at time $t$. $\mathbf{1}_X$ is the indicator function of taking actions, which is 1 when the statement $X$ is true and 0 when $X$ is false. Considering the limited fuel budget, we assume that we have a fixed cost for each action taken by the agent here, and doing nothing brings no cost. Then,

to describe the goal of the game, we define the function of the final status by hasLanded(), which is 0 when not landing; 1 when the agent has landed softly on the landing pad; and $-1$ when the lander runs out of fuel or loses contact with the pad on landing. The reward function rewards the agent for reducing its distance to the landing pad, decreasing its speed to land smoothly and keeping the angular speed at a minimum to prevent rolling. Additionally, it penalises the agent for running out of fuel and deters the agent from taking off again after landing.

By introducing a reward function with minimally bounded costs, our goal was to test if LICRA can exploit the optimal policy. In Fig. 3, we observe that the LICRA agent outperforms all the baselines, both in terms of sample efficiency and average test return (total rewards at each timestep). We also observe that LICRA enables more stable training than PPO, PPO-Lagrangian and CPO.

**Ablation Study 1. Prioritisation of Most Important Actions.** We next tested LICRA's ability to prioritise where it performs actions when the necessity to act varies significantly between states. To test this, we modified the Drive Environment to now consist of a single lane, a start state and a goal state start (at the end) where there is a reward. With no acceleration, the vehicle decreases velocity. To reach the goal, the agent must apply an acceleration $a_t \in [-1, 1]$. Each acceleration $a_t$ incurs a cost $C(a_t)$ as defined above. At zones $k = 1, 2, 3$ of the lane, if the vehicle is travelling below a velocity $v_{min}$, it is penalised by a strictly negative cost $c_k$ where $c_1 < c_2 < c_3$. As shown in Fig. 4, when the intervention cost increases i.e. when $K \to \infty$, LICRA successfully prioritises the highest penalty zones to avoid incurring large costs.

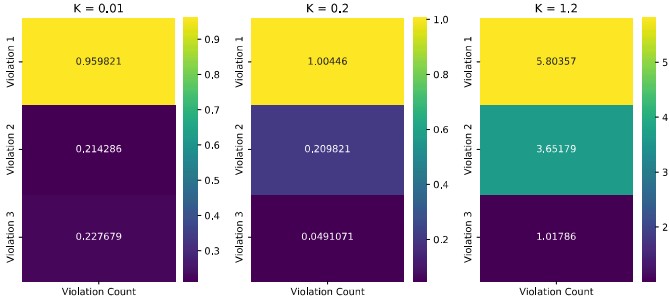

Figure 4: Results for Ablation Study 1. Heatmaps display the number of times the agent drives below $v_{min}$ in the penalty zones. Violation 1 refers to the lowest cost zone, whilst Violation 3 refers to the largest cost zone. $K$ refers to the fixed cost for taking an action. Results averaged over 5 seeds.

## 8 CONCLUSION

We presented a novel method to tackle the problem of learning how to select when to act in addition to learning which actions to execute. Our framework, which is a general tool for tackling problems of this kind seamlessly adopts RL algorithms enabling them to efficiently tackle problems in which the agent must be selective about when it executes actions. This is of fundamental importance in practical settings where performing many actions over the horizon can lead to costs and undermine the service life of machinery. We demonstrated that our solution, LICRA which at its core has a sequential decision structure that first decides whether or not an action ought to be taken under the action policy can solve tasks where the agent faces costs with extreme efficiency as compared to leading reinforcement learning methods. In some tasks, we showed that LICRA is able to solve problems that are unsolvable using current reinforcement learning machinery. We envisage that this framework can serve as the basis extensions to different settings including adversarial training for solving a variety of problems within RL.

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
