# OpenReview forum: "Timing is Everything: Learning to Act Selectively with Costly Actions and Budgetary Constraints"
_ICLR.cc/2023/Conference — ICLR 2023 poster_

### Official Review · Reviewer_KqHR · 2022-10-23

**Confidence:** 3
**Correctness:** 4
**Technical Novelty And Significance:** 2
**Empirical Novelty And Significance:** 3
**Recommendation:** 6

**Clarity, Quality, Novelty And Reproducibility:**

The paper is well written and well executed. While not novel per se, the approach would be of wide practical importance and demonstrating _how_ it works with existing standard RL algorithm is super useful. However unclear if the source code of the experiments will be released.

**Strength And Weaknesses:**

**Strengths**

The paper applies ideas from hierarchical RL to help solve a class of decision making problem of wide practical importance. The idea is intuitively straightforward but well executed with applications to both on-policy and off-policy RL algorithms demonstrated on standard gym environments. Convergence and optimality guarantees are also proven in the case of application to standard Q-learning. The paper also shows how this framing allows them to do well on RL with a budget class of decision making problems.

**Weaknesses**
Calling the framework itself novel is slight overselling because it's standard hierarchical RL framing. Given the improvements in exploration with this framing, the results are not surprising.

**Summary Of The Paper:**

The paper focuses on a subset of decision making problems particularly relevant to real world applications: learning _when to act_ along with the appropriate action to execute. While this problem can be framed as a standard RL problem with no-op action, the paper demonstrates why this is suboptimal. To this end, the paper proposes a new method termed LICRA that can be seamlessly applied to any existing RL algorithm to tackle such problems. The core idea is akin to hierarchical RL. However, limiting to a binary choice in switching policy simplifies many issues that make hierarchical RL difficult to make work in practice. Results are demonstrated on standard gym environments with extensions to standard deepRL algorithms.

**Summary Of The Review:**

Overall even though it's overall an incremental paper, it is well executed and the problem being tackled is of high utility especially in the industry.

---

> ### Author Response · Authors · 2022-11-18
> **Response to Reviewer KqHR**
>
> We thank the reviewer for their careful reading and kind comments.
>
> As you alluded to, the LICRA framework is of practical importance in settings such as robotics and autonomous driving that require efficient methods for solving problems in which actions incur costs (or produce wear & tear). Also, less commonly found in hierarchical RL, we also provide a detailed theoretical analysis including convergence guarantees of our method.
>
> We hope that the Reviewer will voice their appreciation during the internal discussion phase so that the framework can be made use of by the community.
>
> Best wishes,
>
> The Authors.

---

### Official Review · Reviewer_DavQ · 2022-10-23

**Confidence:** 3
**Correctness:** 4
**Technical Novelty And Significance:** 3
**Empirical Novelty And Significance:** Not applicable
**Recommendation:** 6

**Clarity, Quality, Novelty And Reproducibility:**

While the idea of impulse control and balancing costs with optimal actions is no particularly novel idea, the authors proposal to incorporate it into RL methods appears original and is well motivated and clearly laid out.

**Strength And Weaknesses:**

The paper has a clear underlying idea, and motivates it well both intuitively and mathematically.

**Summary Of The Paper:**

The paper proposes a type of policy design for reinforcement learning in the presence of costs to taking actions. The authors propose to learn the decision on whether to act in a specific state separately from learning the action itself. The authors note that this setup reduces the computational complexity and therefore makes policy learning with transaction costs more effective. They further demonstrate how to adjust the method to learn policies with budget constraints by augmenting the state space and evaluate the performance of their methods in benchmark problems.

**Summary Of The Review:**

I think the idea of the paper adds to the existing literature by providing a well-motivated idea on incorporating action costs into policy learning and demonstrates the effect in meaningful experiments.

---

> ### Author Response · Authors · 2022-11-18
> **Response to Reviewer DavQ**
>
> We thank the reviewer for their careful reading and kind comments.
>
> As you alluded to, the purpose of our paper is to enable methods for solving problems with costs for each action to handle unknown environments and adapt reinforcement learning (RL) methods to handle such settings. This is the first paper to do this (to our knowledge). In doing so, our framework, LICRA allows problems within RL settings such as problems within robotics that have action costs to now be efficiently solved.
>
> We hope that the Reviewer will voice their appreciation during the internal discussion phase so that the framework can be made use of by the community.
>
> Best wishes,
>
> The Authors.

---

### Official Review · Reviewer_sGpp · 2022-10-24

**Confidence:** 4
**Correctness:** 4
**Technical Novelty And Significance:** 3
**Empirical Novelty And Significance:** 3
**Recommendation:** 8

**Clarity, Quality, Novelty And Reproducibility:**

The proposed algorithm is simple but clearly shows the benefits of using an impulse policy to learn when to act along with simultaneous learning of a separate policy that decides the action to take. The paper is easy to read and the writing is overall easy to understand, barring the points I have mentioned above in the weaknesses.


**Strength And Weaknesses:**

Strengths:

The experiments are designed to clearly demonstrate the advantages of using a nested optimization approach to efficiently identify the null or no cost action that enables the RL agent to follow a pre-specified intervention or cost budget. Even when there is no cost involved, ablation results in the appendix show that LICRA is able to prioritize learning a policy for those states that ensure higher average returns. Compared to standard RL algorithms like SAC or PPO that fail to converge to an optimal policy when faced with a difficult exploration problem, LICRA consistently converges to a high reward stable solution.


Weaknesses:

1. Sec 5, first paragraph: “..which can be evaluated online therefore allowing the g.” - This sentence seems incomplete.

2. Sec 6, Fig 1: The description of Fig 1 on page 8 says “The agent can act after a time interval of 0.01 seconds and the episode ends after 75 steps.” But the x-axis in Fig 1 continues till 500k timesteps. It would help to clearly explain how to interpret the evaluation process in this setting.

3. Page 8, Driving Environment Fuel Rationing : “2) if accelerations should be performed …” -> “2) accelerations should be …”?

4. Fig 1 / 2 / 3 : The number of random seeds used to get the error bars is not specified.

5. Fig 4: Why does the violation count increase for all cost levels when K is increased? Are the numbers in the heatmap supposed to add to 1 for each K? What is the total time of the trials?


**Summary Of The Paper:**

This paper proposes an algorithm called LICRA to enable an RL agent to learn when to act and which actions to take while optimizing for the incurred action cost. LICRA is inspired by the method of impulse control to learn an impulse policy that decides when to act. LICRA also simultaneously learns the action selection policy that chooses the action to execute if and when the impulse policy decides to act.


**Summary Of The Review:**

I recommend accepting the paper if the comments above are addressed. The stated theorems in the main paper support the proposed formulation of the algorithm and the results also support the main claims made in the paper.

---

> ### Author Response · Authors · 2022-11-16
> **Response to Reviewer sGpp**
>
> We thank the reviewer for the insightful suggestions and comments. We’re also glad the reviewer appreciates our paper and the problem it tackles. We address the reviewer's comments and questions individually and we are uploading an revised version of our paper which now includes modifications, which, where needed, we detail in our responses to the reviewer’s points.
>
> > **Sec 6, Fig 1: Description of Fig 1 on page 8 and the evaluation process in this setting.**
> #### Answer:
> While one episode ends after 75 steps, we train the compared algorithms for much more than one episode. We plot the average return against the total number of timesteps the algorithm has been trained for. In case of the Merton problem each 75 steps translate to one episode of training.
>
> > **I).“Why does the violation count increase for all cost levels when $K$ is increased?” II). The numbers in the heatmap”**
>
> #### Answer:
> I). The intervention cost in this setting is given by $C(a_t)=K+a^2_t$ (see Driving Environment Fuel Rationing section on pg 8) so that larger values of $K$ means that applying an acceleration incurs larger (fixed) costs (this increase is irrespective of the magnitude of the acceleration). In this setting, penalties are incurred whenever the vehicle's speed is below a certain value, specifically a penalty $c_j$ which depends on which zone $j\in\\{1,2,3\\}$ the agent is in. Therefore, for sufficiently large values of $K$, the cost associated to applying an acceleration (to achieve the velocity required to avoid the penalty) is larger than the penalty $c_j$. This means an optimally behaving agent will incur more penalties as $K$ grows large in order to avoid incurring the higher accleration costs.
>
> II). The numbers in the heatmap represent the frequency of violations in each penalty zone. We have updated the script to make these points clearer.
>
> III). In terms of time, these ablation experiments are very simple, due to the single lane and one car, taking at most ~20 minutes for a run.
>
> >**Fig 1, 2, 3 : The number of random seeds used to get the error bars is not specified.**
> #### Answer:
> Thanks for the comment. We have provided the detailed settings in our appendix and now included these details in the corresponding places in paper.
>
>
> >**Incomplete sentence in Sec 5, first paragraph.**
>
> #### Answer: Thanks! We have modified the sentence accordingly in our update.

---

### Official Review · Reviewer_dHaP · 2022-10-27

**Confidence:** 4
**Correctness:** 2
**Technical Novelty And Significance:** 3
**Empirical Novelty And Significance:** 2
**Recommendation:** 5

**Clarity, Quality, Novelty And Reproducibility:**

Quality: As mentioned in the weaknesses above, there are many correctness issues with this paper. In addition I would like to see a comparison to simple baselines that incorporate an inductive bias towards inaction (also discussed in the weaknesses).

Clarity: Overall I found the paper to be well explained and clear (except in the cases where incorrect claims were made which of course led to confusion).

Originality / Novelty: I do not know of other work tackling this setting, and if the approach did significantly outperform a good baseline, I think that would be a significant contribution.

**Strength And Weaknesses:**

Strengths:
- The problem setting is an important one, especially when applying RL to the real world.
- The approach in the paper is principled and elegant, while remaining fairly simple.

Weaknesses:
- It seems plausible to me that the main way in which LICRA provides a benefit is that LICRA provides an inductive bias towards inaction. It would be good to compare to simple baselines that also provide such an inductive bias: for example, use a baseline method like PPO or SAC, but extend the action space with one extra dimension corresponding to “don’t take any action”. This would test whether the extra algorithmic complexity is actually necessary.
- The empirical evaluations of LICRA do not seem that compelling, except in the case of the Drive environment. More worryingly, the authors exaggerate the benefits of LICRA in the text (see specific examples below).
- The authors make another incorrect claim about the asymptotic complexity of LICRA, discussed below.

Incorrect claims made in the paper:

> Conversely, owing to the binary decision space for g (c.f. Prop. 1), LICRA requires |S^c_I| + |S_I||A| evaluations.

This seems impossible. When you only have black-box access to a reward function R(s, a), you at least need to look at each entry in the reward function in order to get a guarantee of convergence to the optimal policy (what if the one entry you haven’t looked at yet is the highest possible reward value?), which implies that any algorithm must have complexity at least Ω(|S||A|). However this is contradicted by the claim above.

Possibly the claim is about the case where you already know the set S_I? But then the size of the set S^c_I is irrelevant, since you always take the null action in that setting. (And also this is not a reasonable measure of the complexity of LICRA, because in an actual setting you don’t know the set S_I.)

This is not the only place where the authors make claims about the complexity benefits of LICRA, but it is the one where the claim is clearly incorrect. Elsewhere the claims are vague enough that they are not obviously false: for example the authors also say “By isolating the decision of whether to act or not, the LICRA framework may also reduce the computational complexity in this setting.” Nonetheless I think such statements should be removed from the paper given that they are not supported by any of the evidence presented in the paper.

> The results of training are shown in Fig. 1 which clearly demonstrates that LICRA_PPO finds a better policy than standard PPO.

Looking at Figure 1 it does not seem obvious that LICRA_PPO outperforms standard PPO; overall they seem to be similar. LICRA_PPO does learn a good policy faster than PPO, but this is a different claim.

> Also comparing the variance among different seeds, we can see that LICRA_PPO is a much more stable algorithm than the other two.

(This is also referring to Figure 1.) This claim also looks false based on Figure 1 and I am confused what the authors are seeing that I am not. As far as I can tell the blue error bars of LICRA_PPO are if anything _larger_ than the orange error bars of PPO.

> In Fig. 3, we observe that the LICRA agent outperforms all the baselines, both in terms of sample efficiency and average test return (total rewards at each timestep).

Looking at Figure 3, I agree that LICRA-SAC outperforms the baselines (at least on average test return), but LICRA-PPO seems comparable with SAC and PPO.

Minor issues:

In the preliminaries, right after equation (1), you say:

> \mathcal{R}(s, a) = R(s, a) 1_{a∈A/{0}} + R(s, 0)(1−1_{a∈A/{0}})

Isn’t this just equivalent to saying \mathcal{R}(s, a) = R(s, a)?

In the definition of the intervention operator M, policy \pi is associated with M and policy \pi’ is associated with Q, but in the next sentence explaining the interpretation these are switched. Please choose a consistent usage. (Personally I find it more intuitive to have \pi’ associated with M, since I interpret \pi’ as the intervention policy.) I would also recommend using parentheses to clarify scopes (in particular I was initially unclear on the scope of the a_{\tau_k} variable).

Just before equation (3), you say:

> Denote by M[Qπ,g] the intervention operator acting on Qπ,g when the immediate action is chosen according to an epsilon-greedy policy.

Shouldn’t this be a fully greedy policy, rather than epsilon-greedy, in order to satisfy the Bellman equation at optimality? (It seems like this would be needed for Theorem 1, for example.)

Typo: envisge → envisage

**Summary Of The Paper:**

In many situations an RL agent incurs costs whenever it acts. Simply applying standard RL algorithms in such situations often doesn’t work, because it is not easy for the algorithm to choose “no action” instead of “small action” (particularly in continuous action spaces). To deal with this, the authors propose Learnable Impulse Control Reinforcement Algorithm (LICRA), which learns a two-step policy: the first step decides whether or not to act, and the second step decides what action to take (if any is needed). The authors provide:

1. An algorithm to learn the two-step policy from experience,
2. A proof of convergence to the optimal policy in the tabular and linear function approximator settings,
3. An application to the setting in which there is a fixed budget of actions, and
4. An empirical evaluation of LICRA on three environments with costly actions.

**Summary Of The Review:**

While the problem setting is of interest and the algorithm proposed is simple and theoretically justified (if not particularly surprising), I would recommend rejecting, for two main reasons:

1. It seems plausible that the benefits come from an inductive bias towards inaction, but this can also be achieved much more simply by changing the network architecture to add a direct prediction for “take no action” and then using standard RL algorithms. This should be a baseline to which LICRA is compared.
2. The paper contains several incorrect claims, most worryingly claims that overstate the benefits of their method in the empirical settings they evaluate in.

UPDATE: The author response has addressed many of my concerns (particularly the ones about incorrect claims). My first worry remains. As a result I am increasing my score by one step.

---

> ### Author Response · Authors · 2022-11-15
> **Author response to Reviewer dHaP**
>
> >> *Comparison against baselines* : I) The reviewer’s suggestion to compare against standard baselines with added “take no action”. II) "It seems plausible that the benefits come from an inductive bias towards inaction”
>
> #### Answer: I) The reviewer has suggested that we compare against a baseline which is constructed by adding a direct prediction for “take no action” and then using standard RL algorithms. We emphasise that all the baselines have access to the action “$0$” which incurs $0$ costs and exerts no influence on the transition dynamics - therefore the reviewer’s point about extending the action space of the baselines to include “don’t take any action” is already addressed in the current version of the paper. Please see the first paragraph of Section 7.
>
> #### Answer: II) We clarify that LICRA does not introduce a bias of any kind (i.e. it does not maintain a tendancy towards inaction). The purpose of the high level policy is to determine which set of states to act which it does so when it is profitable.

---

> > ### Comment · Reviewer_dHaP · 2022-11-27
> > **This misunderstands my point**
> >
> > I agree that in discrete environments (Merton and Lunar Lander) you are already comparing to the baseline I mention. Sorry for not saying that in the review. However, this is not the case in the continuous setting (the Drive environment), and that is where you see the strongest performance.
> >
> > I assume that in the Drive environment you are using a function approximator to produce a continuous scalar in the range [-1, 1]. (Incidentally, please include details on the architecture of the function approximator used for all the experiments, for the sake of reproducibility.) While it is technically possible for the function approximator to output 0, most architectures (e.g. a single-layer or multi-layer perceptron) would not output 0 in practice even if trained to do so (instead outputting, say, 0.0001). This means that in this environment the baselines are always going to incur a fixed cost of at least $K$ on each timestep.
> >
> > (Indeed, you say this in the paper itself: "We believe one reason for the success of LICRA is that it is far easier for it to utilise the null action of zero acceleration/braking than the other algorithms, whilst all the algorithms have a guaranteed cost at every time step whilst not gaining a sizeable reward to counter the cost.")
> >
> > In such an environment, the first obvious step to take for a meaningful baseline is to give the agent some way of actually choosing action 0. My recommendation would be to have the function approximator provide a vector of two numbers $(x_1, x_2)$, and then have the environment action be given by $a = 0$ if $x_1 < 0$ else $x_2$. This can then be trained with PPO or SAC as normal, and provides a much more meaningful baseline.

---

> > > ### Author Response · Authors · 2022-11-29
> > > **Thanks for the clarification + Our Response**
> > >
> > > Thanks a lot for the clarification and detailing your interesting thoughts for constructing a new baseline.
> > >
> > > We would argue that the Reviewer's construction yields a degenerate variant of LICRA.
> > >
> > > To explain, by Prop. 1, in LICRA's Q-learning variant, the (high level) intervention policy $\mathfrak{g}$ is characterised by the expression: $\mathfrak{g}(s)=H((\mathcal{M}^{\pi}Q-Q)(s,a))$ where $H$ is the Heaviside function i.e. for any $y\in\mathbb{R}$, $H(y)= 0$ when $y\leq 0$ and $H(y)=1$ otherwise and the optimal intervention policy $\mathfrak{g}^\star$ is given by $\mathfrak{g}^\star=H(\mathcal{M}^{\pi^\star}Q^\star-Q^\star)$ where $\pi^\star$ is a greedy policy. Defining the variable $x_1$ by $x_1\equiv (\mathcal{M}^{\pi}Q-Q)(s,a)$ means that in LICRA, the agent intervenes only when $x_1\geq 0$ while the action policy $\pi$ determines the action selection when activated. With this, we can see that the baseline suggested by the Reviewer resembles a (degenerate) variant of LICRA where $x_1$ is now being instead learned as an output to a single policy (function approximator) architecture that encapsulates both LICRA's action policy $\pi$ and the intervention policy $\mathfrak{g}$ (our thought here is that such a setup would lose some of the efficiency and scaling benefits of LICRA's distributed/hierachical RL architecture in which $\pi^\star$ is learned separately and communicates its proposals to $\mathfrak{g}$).
> > >
> > > We are grateful to the Reviewer for the interesting suggestion which we think would compliment the current set of comparisons nicely. We are running experiments to include this comparison now. As to our knowledge no current methods have this setup (and given the observations above), we'll include this study as an Ablation Study of the LICRA framework.
> > >
> > > Also with regard to the architecture details, thanks for pointing this out. We have now also updated the paper to include details on the architecture of the function approximator for all our experiments.

---

> > > > ### Comment · Reviewer_dHaP · 2022-11-29
> > > > **I agree**
> > > >
> > > > Yes, I agree that this construction can be thought of as a different way of learning the intervention policy. I'm interested in it as a baseline because it is easier to implement (just a very simple action space wrapper) and doesn't require a change to the learning algorithm.

---

> > > > > ### Author Response · Authors · 2022-11-29
> > > > > **We agree that this is an interesting construction**
> > > > >
> > > > > We agree that this is an interesting construction and will add this variant to our Ablation studies (Openreview doesn't allow us to upload updates at this moment).
> > > > >
> > > > > We would argue that given the efficiency and scalability benefits of distributed/multi-agent and hierachical architectures (of which LICRA is) in settings similar to the one we consider, a single policy variant would likely have poorer scalability and efficiency.

---

> ### Author Response · Authors · 2022-11-15
> **Author response to Reviewer dHap**
>
> >**Our claims about the benefits over the baselines**
>
> #### Answer: We firstly emphasise that as LICRA is a plug \& play enhancement tool, the most relevant baselines to compare to are the corresponding base learners (e.g. comparing LICRA_SAC to SAC).
>
> In the more complex environments, namely Lunar Lander and Drive Environment, LICRA noticeably outperforms the corresponding baselines (in Lunar Lander LICRA_SAC yields a 60% gain over SAC and LICRA_PPO yields an 18% gain over PPO. In Drive Env. LICRA_SAC yields a 372% gain over SAC and LICRA_PPO yields an 996% gain over PPO) .
>
> The Merton problem has one-dimensional state space and action spaces making it by far the simplest setting (we chose it because of its easy exposition and widespread application). Even in this simple setting, LICRA on top of PPO (LICRA_PPO) solves the problem noticeablty faster than plain PPO requiring  around 30k training steps fewer than plain PPO. We have now corrected the statement in the paper to read that LICRA_PPO enables the optimal policy to be found *more quickly* than standard PPO which is in-keeping with the claim of efficient learning we made in the paper (please see Sec. 7, pg 7-8).
>
> We have now added the following final scores to the captions that clearly indicate LICRA's superior empirical performance relative to the base learner:
>
> Merton Problem (final scores): LICRA_PPO(26.40), PPO(19.60)
> LunarLander (final scores): LICRA_SAC(98.03), SAC(61.20), LICRA_PPO(58.84), PPO(49.77).
> Drive Env(final scores): LICRA_SAC(1.75), SAC(0.37), LICRA_PPO(2.63), PPO(0.24).

---

> ### Author Response · Authors · 2022-11-15
> **Author response to Reviewer dHap**
>
> >> **Claim about the asymptotic complexity of LICRA and learning speed**
>
> #### Answer:  We thank the reviewer for their comment about our learning speed claims. Our main goal with the calculation in Sec. 4 was to merely motivate why LICRA can improve learning speed. We wish to point out to the reviewer that having learned $\mathcal{S}_I$, the agent wouldn't not act everywhere as the reviewer has stated but rather, the agent would need to **i)** act (only) at all states in $\mathcal{S}_I$ **ii)** determine the optimal action for all states at $\mathcal{S}_I$. Note also that the setting we consider is a continuous action setting.
>
> Unlike the discrete action setting, in the continuous action setting, learning optimal actions does not involve performing evaluations of all reward function entries. Since each action incurs a cost of at least $c>0$, acting at some states outside of $\mathcal{S}_I$ may produce an appreciable reduction in expected return i.e $Q(s,a\equiv 0)\gg Q(s,a')$ for any $a'\in\mathcal{A}/\\{0\\}$. With this, using our binary policy framework, we suggest (and the empirical results of Sec. 11.2 suggest) that in such states, LICRA can quickly learn to take the null or $0$ action i.e. find the optimal binary policy since all action-policy proposals of $a'\in\mathcal{A}/\\{0\\}$ to the binary policy would lead to a lower payoff. Given this, these benefits accumulate whenever $\mathcal{S}_I$ is relatively small which is what we observed empirically as shown in Sec. 11.2. For ease of exposition, we gave this motivation in the discrete action space setting even though this isn't the setting we are concerned with in general. **As this has obstructed clarity, in line with the reviewer's suggestion, we have now removed this claim from the paper.**
>
> We have updated the script to highlight where we feel this claim is partially supported by empirical results, namely Sec 11.2 of the appendix. The support it provides is due to that fact that as our calculations suggest, the benefit is biggest when the intervention region is small --- in keeping with the results of our learning speed claim.

---

> ### Author Response · Authors · 2022-11-15
> **Author response to Reviewer dHap**
>
> >>  **Our stability claim.**
>
> #### Answer:
> We thank the reviewer for this comment. We have run more seeds (up to a total of 20) on the Merton problem and reported the results in the updated version of the paper. We can compare the average variance between steps for both algorithms, which yields the following:
> PPO (9.75), LICRA_PPO (9.19). While we have observed that can LICRA decrease the variance of returns between seeds during training, we admit that this decrease might be considered too small to safely say it is significant, variance reduction is also not a proposed central benefit of LICRA. We have now removed this comment from the paper.

---

> ### Author Response · Authors · 2022-11-15
> **Author response to Reviewer dHap**
>
> > **Definition of $\mathcal{R}(s, a):= R(s, a) 1_{a∈A/\\{0\\}} + R(s, 0)(1−1_{a∈A/\\{0\\}})$.**
>
> #### Answer:
> We included this notation so as to include the case where the agent has a "null action" and this is not an element within the action space i.e. it represents simply not taking any action. The other case of a null action (which features in our experiments) is the action when $a\equiv 0$.

---

> ### Author Response · Authors · 2022-11-15
> **Author response to Reviewer dHap**
>
> > **The intervention operator with an epsilon-greedy policy.**
>
> #### Answer:
> We thank the reviewer for this comment. Provided epsilon is a random variable with $0$ expectation (which is normally assumed) our proofs are valid for an epsilon greedy regime.

---

> > ### Comment · Reviewer_dHaP · 2022-11-27
> > **Still doesn't seem right to me**
> >
> > (Sorry for the bad Latex here; this editor's Latex support is terrible.)
> >
> > Your current definition of M is:
> >
> > $$\mathcal{M}v(s) = \sup_{a \in A} R(s, a) - c(s, a) + \gamma \sum_{s' \in S} P(s' ; a, s) v(s')$$
> >
> > I assume that to generalize this to an epsilon greedy regime, you would want to have a definition of the form
> >
> > $$\mathcal{M}v(s) = \sup_{a \in A} E_{\epsilon} [ R(s, a + \epsilon) - c(s, a + \epsilon) + \gamma \sum_{s' \in S} P(s' ; a + \epsilon, s) v(s') ] $$
> >
> > Where $\epsilon$ is sampled from a distribution with expectation 0.
> >
> > I don't think this works. For example, consider a very simple system, where the state $s$ is a scalar representing distance on the number line, the action $a$ is a scalar saying how far to travel, so that the next state $s' = s + a$. The reward is given by $R(s, a) = a$, that is the agent is rewarded for moving towards positive values. The cost is given by $c(s, a) = 1000 \cdot (a - 0.1)^2$, that is, the agent is heavily penalized for moving at any speed other than 0.1. (If you want the cost to be 0 when $a = 0$, then imagine that the cost is $c(s, a) = 1000 \cdot a \cdot (a - 0.1)^2$.)
> >
> > Now assume that our initial policy pair always chooses not to act, that is $g(s, x) = 1[x = 0]$. (If you need a policy too, then we can set $\pi(s, a) = 1[a = 0]$, but I think that doesn't matter.) This gives us an initial value function of $v(s) = 0$.
> >
> > If we sample $\epsilon$ from a standard normal, that is, $\epsilon \sim \mathcal{N}(0, 1)$ (which has expectation zero), then:
> >
> > $\mathcal{M}v(s) = \sup_{a \in A} E_{\epsilon} [ R(s, a + \epsilon) - c(s, a + \epsilon) + \gamma \sum_{s' \in S} P(s' ; a + \epsilon, s) v(s') ]$
> >
> > $\mathcal{M}v(s) = \sup_{a \in A} E_{\epsilon} [ R(s, a + \epsilon) - c(s, a + \epsilon) ]$ (since $v(s') = 0$)
> >
> > $\mathcal{M}v(s) < 0$ (because the large magnitude of $\epsilon$ means that the expected cost is much much larger than the expected reward).
> >
> > If we then plug this into Equation (3), we get:
> >
> > $Tv(s) = \max(\mathcal{M}v(s), \mathcal{R}(s, 0) + \gamma \sum_{s' \in S} P(s' ; 0, s) v(s') )$
> >
> > $Tv(s) = \max(\mathcal{M}v(s), 0)$
> >
> > $Tv(s) = 0$ (since $\mathcal{M}v(s) < 0$).
> >
> > So in this environment, with the initial $v(s) = 0$ corresponding to never acting, we have $\lim_{k \rightarrow \infty}T^k v = 0$, which does not correspond to the optimal policy (which corresponds to always choosing $a = 0.1 + c$ for some small $c$ that I'm not bothering to calculate), making it a counterexample to Theorem 1.

---

> > > ### Author Response · Authors · 2022-11-29
> > > **Response to the Reviewer**
> > >
> > > Thanks for the insightful example. We believe the Reviewer's example highlights the fact that in cases where the policy must maintain some level of stochasticity in its output (even after training), the optimal solution of an MDP can differ from that when the policy space includes policies that can execute purely greedy actions [1].
> > >
> > > We have updated the script to include the condition that our policy space must satisfy the condition of containing policies that are greedy in the limit of infinite exploration (GLIE).  This is a standard condition required to guarantee the convergence of many RL algorithms [2].
> > >
> > > We believe this resolves your concern and look forward to your response.
> > >
> > > [1] Korn, Ralf. "Optimal impulse control when control actions have random consequences." Mathematics of Operations Research 22.3 (1997): 639-667.
> > > [2] Singh, S. P., Jaakkola, T., Littman, M. L., and Szepesv´ari, C. (2000). Convergence results for single-step on-policy reinforcement-learning algorithms. Machine Learning, 38(3):287–308.

---

> > > > ### Comment · Reviewer_dHaP · 2022-11-29
> > > > **Policies can't be GLIE**
> > > >
> > > > > We have updated the script to include the condition that our policy space must satisfy the condition of containing policies that are greedy in the limit of infinite exploration (GLIE).
> > > >
> > > > I am a little confused what this means. GLIE is a condition on learning algorithms, whereas policy space is a set of distributions over actions given states. I don't think policies can be GLIE; that sounds like a type error.
> > > >
> > > > Also, Theorem 1 doesn't have anything to do with learning dynamics, since it is stating a property of the defined Bellman operator, rather than a learning algorithm. Similarly I assumed Theorem 2 is talking about a version of Q-learning in which exploration is not an issue because you simply iterate over all states and actions, though upon rereading maybe that's wrong.
> > > >
> > > > I still think that, at least for Theorem 1, all you need is for $\mathcal{M}$ to intervene with a greedy policy -- and indeed that's what your appendix does. (I'm now a bit more confused what Theorem 2 is saying but if it is the version where we ignore exploration then that should work for Theorem 2 as well.)

---

> > > > > ### Author Response · Authors · 2022-11-29
> > > > > **We agree**
> > > > >
> > > > > Thanks a lot for the comment.
> > > > >
> > > > > We agree with the Reviewer's suggestion and we'll update the Theorem to specify that we consider only greedy policies (as the Reviewer points out, this is sufficient).
> > > > >
> > > > > We also agree with the Reviewer that the GLIE condition applies to RL algorithms. Our thinking behind adding the GLIE condition on the base RL algorithms (recalling that LICRA is plug & play) is to enable our convergence results to also cover RL algorithms (as base learners) that execute random exploratory behaviour during their training phase.
> > > > >
> > > > > In our update we will specialise to the greedy case for all our convergence analyses which we agree will resolve this issue.

---

> ### Author Response · Authors · 2022-11-15
> **Author response to Reviewer dHap**
>
> We thank the reviewer for the insightful suggestions. We address the reviewer's comments and questions individually, we have also made some adjustments to the script which we refer the reviewer to in our responses below.

---

> ### Author Response · Authors · 2022-11-18
> **Did we address your concerns?**
>
> Dear reviewer dHaP,
>
> We are grateful for your review of our paper.
>
> We have sought to address each of your points in detail during our rebuttal. We believe all the points you have raised have been entirely addressed by our rebuttal but we have not received a response from you during the rebuttal period.
>
> We would be grateful if you could confirm our responses have resolved all of your comments or let us know if any points of yours remain unaddressed.
>
> Best wishes,
>
> The authors

---

> > ### Comment · Reviewer_dHaP · 2022-11-27
> > **Sorry for the late response.**
> >
> > Thanks for the rebuttal! I believe it has addressed most of my points, except the one about an epsilon-greedy policy and the one about baselines, for which I've written more detailed responses. As a result I will raise my score one step.

---

> > > ### Author Response · Authors · 2022-11-29
> > > **Thanks for the response.**
> > >
> > > Thanks a lot for the comments and questions. We’ve replied to your comments individually below.

---

### Decision · Program_Chairs · 2023-01-20

**Decision:**

Accept: poster

**Justification For Why Not Higher Score:**

- In continuous action settings, it's not clear if the success of the method is simply because "it is far easier for it to utilise the null action of zero acceleration/braking than the other algorithms". A simple baseline of including a discrete "null" action would help provide more insight to where the gains are coming from. Though, it is good that the authors seem to be open to adding ablations to this experiment, and the discrete action experiments do show improvements.
- The experimental problem domains are all rather simplistic/toy. It's not clear if experiments on three simple domains, one of which does not show a particularly convincing improvement, meets the bar for ICLR. Experiments on larger scale and/or real problems (and also just more problem domains) would make the impact of the method even more convincing.


**Justification For Why Not Lower Score:**

- The method is a clever idea that is generally well-done and well explained.
- The experiments illustrate the benefit of the key idea.

**Metareview: Summary, Strengths And Weaknesses:**

Below are the pros and cons of the paper, which the reviewers in the discussion meeting all agreed on. This paper is generally borderline and no reviewer had strong opinions about acceptance or rejection,

Pros:
- The method is a clever idea that is generally well-done and well explained.
- The experiments illustrate the benefit of the key idea.

Cons:
- In continuous action settings, it's not clear if the success of the method is simply because "it is far easier for it to utilise the null action of zero acceleration/braking than the other algorithms". A simple baseline of including a discrete "null" action would help provide more insight to where the gains are coming from. Though, it is good that the authors seem to be open to adding ablations to this experiment, and the discrete action experiments do show improvements.
- At least one reviewer suspects that they would not use this method if they encountered the problem studied in this paper and instead opt for the simple baseline above in continuous action settings. Though, experiments on the setting might alleviate that concern if the method showed strong improvements.
- The experimental problem domains are all rather simplistic/toy. It's not clear if experiments on three simple domains, one of which does not show a particularly convincing improvement, meets the bar for ICLR. Experiments on larger scale and/or real problems (and also just more problem domains) would make the impact of the method even more convincing.

As a side note, the lunar lander results improvements look quite small at first glance, but when looking more closely at the y-axis, the reviewers & I realized that the improvements are larger than they first appear. I might recommend to zooming in on the y-axis, e.g. starting it at -75, so that it is easier to see the differences.

**Note From Pc:**

if the above contains the word "oral" or "spotlight" please see: "oral" presentation means -> notable-top-5% and "spotlight" means -> notable-top-25%. As stated in our emails, we are disassociating presentation type from AC recommendations

**Summary Of Ac-Reviewer Meeting:**

Reviewer KqHR did not reply to the two emails to schedule the meeting, so only three reviewers attended the meeting. We discussed the methods section, the experiments, and the high-level pros and cons of the paper.